

# Farming practices affect the amino acid profiles of the aquaculture Chinese mitten crab

Qingfei Zeng[1], Yuxia Xu[2], Erik Jeppesen[3,4,5], Xiaohong Gu[1], Zhigang Mao[1] and Huihui Chen[1]

[1] State Key Laboratory of Lake Science and Environment, Nanjing Institute of Geography and Limnology, Chinese Academy of Sciences, Nanjing, China
[2] Key Lab of Disaster Monitoring and Mechanism Simulating of Shaanxi Province, Baoji University of Arts and Sciences, Baoji, China
[3] Sino-Danish Centre for Education and Research, University of Chinese Academy of Sciences, Beijing, China
[4] Limnology Laboratory, Department of Biological Sciences and Centre for Ecosystem Research and Implementation, Middle East Technical University, Ankara, Turkey
[5] Department of Bioscience, Aarhus University, Silkeborg, Denmark

Corresponding author
Qingfei Zeng, qfzeng@niglas.ac.cn

## ABSTRACT

Farming operation and amino acid profiles of pond-reared Chinese mitten crabs, *Eriocheir sinensis* (Milne Edwards, 1853), collected from different areas in Jiangsu Province, China were investigated and compared with the aim to elucidate how farming practices affected the nutritional values of three edible tissues (muscle, hepatopancreas and gonad) of crab. The crab pond aquaculture practices including snail input, macrophytes coverage, total commercial feed, the ratio of trash fish to total feed, were much higher in Gaochun and Jintan than that in other sites (having larger pond size), which leads to higher average individual body weight and commercial yields. Further, the mean body weight, muscle weight, carapace length and width, and the ratio of gonad to hepatopancreas were significantly higher in Jintan, Suzhou and Gaochun areas than in other regions. Amino acid assessment showed that all crabs collected delivered high-quality protein (Amino acid score >1 except Valine), the main amino acids being glutamic acid, aspartic acid, and alanine. Significant differences in amino acid profiles were observed between sites, tissues and sexes. Muscles were rich in total amino acids, essential amino acids, and delicious amino acids, followed by gonads and hepatopancreas. The contents of essential amino acids in crab muscles from Gaochun, Jintan, Suzhou and Guannan were significantly higher than those from Suqian, Sihong and Xinghua. All the amino acids except Serine and Glycine were significantly higher in gonads from males than from females. The redundancy analysis revealed that the snail input, trash fish ratio to the total feed, macrophytes coverage and total trash fish supply explained 84.3% of the variation in the amino acid content and structure in crabs from Gaochun, Jintan and Suzhou. Overall, our results show that mitten crabs collected in Jiangsu province had good nutritional quality suitable for human dietary needs, and that farming practices, especially degree of fish-source protein feeding, influence the amino acids composition of crabs.

# INTRODUCTION

Chinese mitten crab, *Eriocheir sinensis* (Milne Edwards, 1853), is a popular food source among Chinese, Japanese, Korean and Southeast Asian consumers because of its delicious taste, unique and pleasant aroma, and nutritional quality (*Wang et al., 2016c*). In the recent two decades, mitten crab aquaculture has considerably grown in China, reaching a production of $7.6 \times 10^8$ kg year$^{-1}$ in 2018, equal to a value of $7.8 \times 10^9$ according to the China Fishery Statistics Yearbook (*CFSY, 2019*). The price of the crabs is generally decided by the individual's weight/size; the larger the crab, the higher the price (*Chen, 2008*; *Wang et al., 2018*). Farming practice affect the taste, nutritional quality and health benefits of cultured species (*Latuihamallo, Iriana & Apituley, 2015*; *Wu et al., 2020a*). Several studies have reported differences in nutrition quality of wild and farmed crabs (*He et al., 2017*; *Kong et al., 2012*; *Wu et al., 2020a*). The comparison between pond cultured crabs based on different farming practices has rarely been reported.

Jiangsu province is the most important farming area in China, contributing 47.2% of total production in 2018. The mitten crab farming is considered to be relatively more environment-friendly in comparison to the aquaculture of Chinese major carps. Farmers transplant aquatic macrophytes to regulate water quality. The farming practices, including daily feed, variation in stocking density, macrophytes transplanting, snail input, differed significantly between farming areas, especially between the south area and north area in the province (*Wang et al., 2016a*). Several studies have reported that environmental and dietary differences affect the protein content and amino acid profiles (*Kasozi et al., 2019*; *Hussain et al., 2018*; *Liu et al., 2018*). Wild-caught crabs, such as swimming crab, mud crab, and Chinese mitten crab, exhibit better nutritional value than cultured crabs (*Liu et al., 2018*). However, there have been few studies focused on the nutrition quality of cultured crabs subjected to different farming practices.

Amino acids are essential nutrients for human growth, physiology, biochemistry and immunity (*Liu et al., 2018*, *2019*). Some amino acids give certain flavors in the crab, such as aspartic acid (umami), threonine (sweet), and arginine (bitter/sweet) (*Kong et al., 2012*). These amino acids can increase attractiveness of the products for the customers. Therefore, study on the composition of amino acids of mitten crab tissues is of key importance in assessing the nutritional and flavor quality of the crab. However, there are no detailed information available about how the farming pattern affects the nutritional quality of pond-reared crabs and why the flavor differs between farming areas, that otherwise could help guiding consumers to choose their favorite food and farmers to optimize farming practices.

We investigated the amino acids composition in three edible tissues of both sexes to elucidate to what extent the nutritional value of mitten crabs is affected by farming patterns. We expected significant differences in amino acids compositions of mitten crabs due to differences in feeding praxis between southern and northern Jiangsu Province (Taking the Yangtze River as the boundary between north and south area).
## METHODS AND MATERIALS

### Schematic overview of study

Figure 1 shows the schematic overview of the study. Herein, both sample collection and farming practices survey was carried out at the same time when the crabs were harvested. Then the crabs were dissected to three edible tissues of both sexes for further growth performance and amino acid analysis. The drivers of amino acids composition of mitten crabs cultured in different farming patterns using a redundancy analysis (RDA).

### Sample preparation

The mitten crabs were sampled in November 2018 from seven major aquaculture areas in Jiangsu Province, China (Fig. 2). Ten intact crabs, 5 males and 5 females, were randomly selected in each pond (2 ponds at each sampling area) and transported on ice alive to the laboratory. The body weight and carapace dimensions (carapace length (CL) and carapace width (CW)) were measured for each individual. Crabs were dissected, and their muscles, hepatopancreas and gonads were weighted and freeze-dried for later further analysis. Since the hepatopancreas and gonads from a single crab had relatively low total weight, the tissue from 5 crabs of each sex per pond were pooled to form for amino analysis. All analyses were repeated twice at each culture area.

### Farming practices survey

The farming practices survey was conducted from farmer during crab sampling (2–4 ponds at each sampling area), including the pond size, stock density, feed composition, macrophytes coverage, and yield.

### Growth performance

The hepatopancreas index (HSI), gonadosomatic index (GSI), the ratio of gonad to hepatopancreas (GH ratio), and condition factor (CF) of the crabs were calculated using the following Eqs. (1)–(4), respectively:

$$HSI(\%) = \frac{\text{hepatopancreas wet weight}}{\text{body weight}} \times 100 \tag{1}$$

$$GSI(\%) = \frac{\text{gonad wet weight}}{\text{body weight}} \times 100 \tag{2}$$

$$GH = \frac{\text{gonad wet weight}}{\text{hepatopancreas wet weight}} \tag{3}$$

$$CF = \frac{\text{body weight}}{\text{carapace length}^3} \tag{4}$$

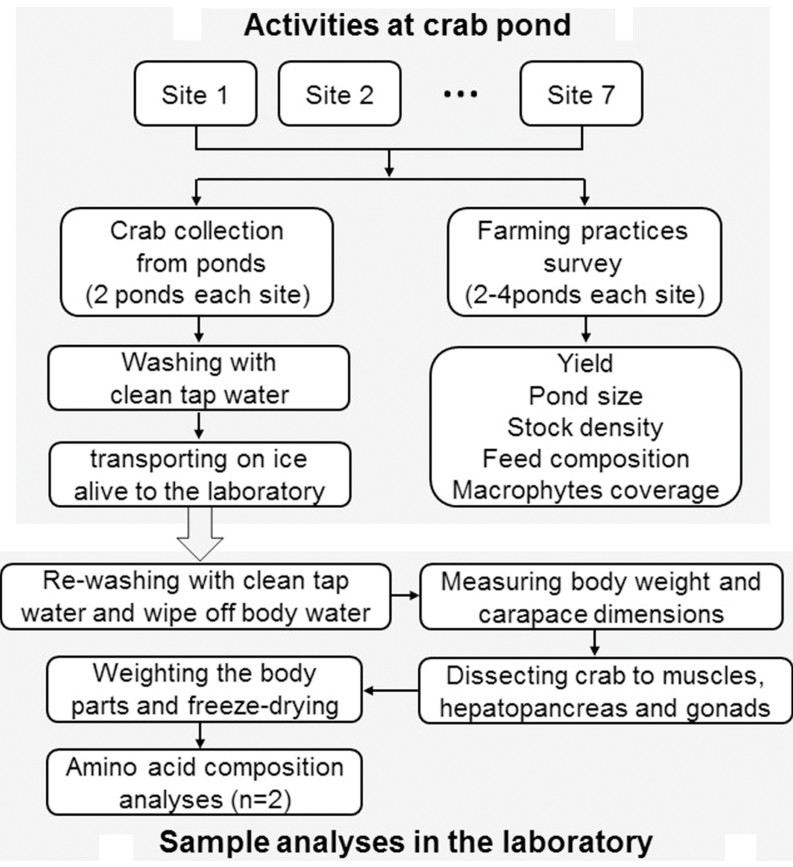

**Figure 1 A schematic overview of the study.** $n$ = number of sample replicates for amino acid analysis.

## Amino acid profiles

The analysis was conducted using an automatic amino acid analyzer (L-8900; Hitachi, Tokyo, Japan) following *Chen, Zhang & Shrestha (2007)*. Duplicated samples were hydrolyzed with 6 mol $L^{-1}$ HCl for 24 h at 110 °C. Hydrolysates were then dissolved in distilled water to a total of 100 mL. A total of one mL of hydrolysate was freeze-dried under vacuum to remove water and HCl. Then, two mL of 0.02 mol $L^{-1}$ HCl was supplemented to solution. One mL of solution filtered through 0.22 μm pore size membrane was used for amino acid analysis. The amino acid profiles were detected by comparing the retention time and peaks with those of amino acid standard (EZChrom Elite for Hitachi Version 3.1).

## Nutritional evaluation

The nutritional evaluation was carried out according to the guidelines of the Food and Agriculture Organization (FAO) for amino acid score and egg protein amino acid score (*FAO/WHO, 1991*). The Eqs. (5)–(7) for amino acid score (AAS), chemical score (CS), and essential amino acid index (EAAI) are as follows:

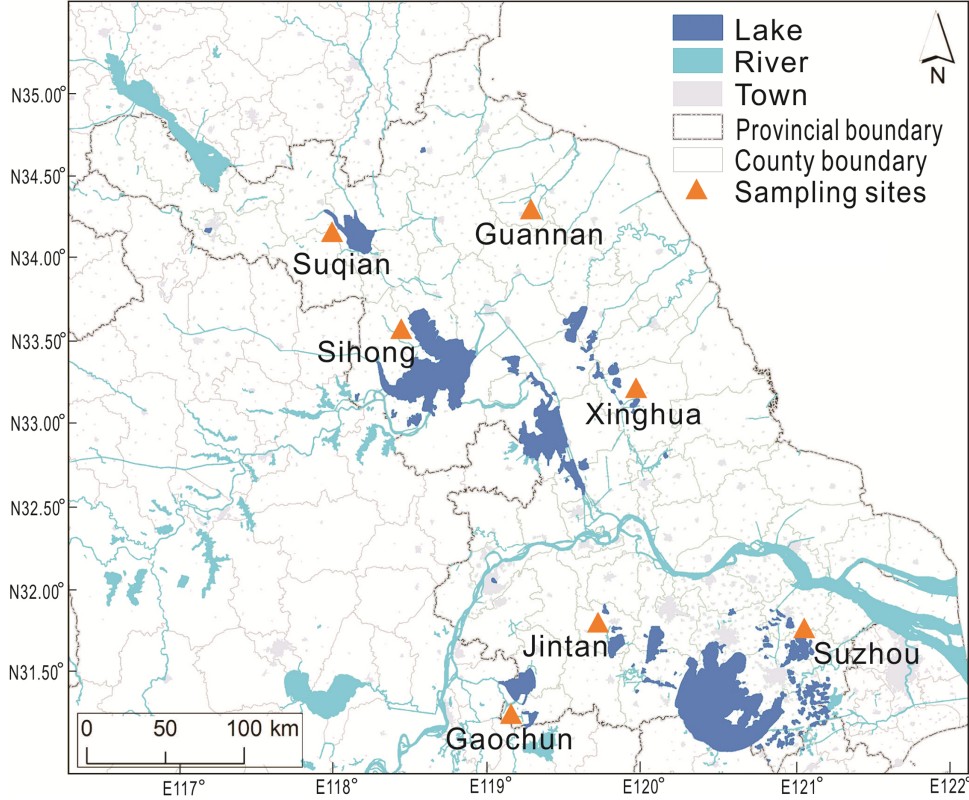

**Figure 2 The sampling sites of Chinese mitten crab (*Eriocheir sinensis* Milne Edwards, 1853) in Jiangsu Province.**

$$AAS(\%) = \frac{\text{sample amino acid content (\%, dry)}}{\text{amino acid content in FAO reference (\%, dry)}} \times 100 \qquad (5)$$

$$CS(\%) = \frac{\text{sample amino acid content (\%, dry)}}{\text{amino acid content in egg protein (\%, dry)}} \times 100 \qquad (6)$$

$$EAAI = \sqrt[n]{\frac{100a}{a_e} \times \frac{100b}{b_e} \times \ldots \times \frac{100h}{h_e}} \qquad (7)$$

where $n$ is the number of essential amino acids compared, a–h are the levels of essential amino acids in the experiment (%, dry), and $a_e$–$h_e$ are the levels of amino acids in the egg protein (%, dry).

## Statistical analysis

All data were shown as mean ± standard error (SE). Homogeneity of variance of data was tested with Levene's test. Two-way analysis of variance (ANONA) was performed to analyze the differences in amino acid profile and growth performance for the effect of sex (♀, ♂) and location (7 sites) using software SPSS 18.0. Alpha level of $p < 0.05$ was accepted as a measure of significant intergroup difference. When the overall treatment effect was

significantly different, Duncan's multiple range test was conducted to compare the means of the individual treatments.

We investigated drivers of amino acids composition of mitten crabs cultured in different farming patterns using a RDA in software CANOCO 4.5. RDA is a direct gradient ordination analysis which summarizes the linear relationship between a multivariate set of response variables to explanatory variables. To investigate the spatial variation of amino acids among the sampling sites, non-metric multidimensional scaling (NMDS) was carried out using PAST 3.0 according to Bray–Curtis similarities using standardized and square-root converted data. Analysis of similarities (ANOSIM) was conducted to test statistically whether there is a significant difference in amino acid profiles among the locations.

# RESULTS

## Farming practice

The farming practices of Chinese mitten crab is shown in Fig. 3. Coin-sized mitten crabs (generally 5.5–7.0 g per individual) were usually chosen for culture at a density ranged from $1.5 \times 10^4$ to $2.9 \times 10^4$ individual ha$^{-1}$ (Fig. 3A). The pond size in Suqian, Sihong, Gaochun and Jintan was around 1 ha, being much lower than that in Guannan (20 ha) and Xinghua area (8.53 ha) (Fig. 3B). The snail input (*Bellamya aeruginosa*, Reeve, 1863) in the pond ranged from 0 to $12 \times 10^3$ kg ha$^{-1}$ (Fig. 3C) and macrophyte (mainly *Elodea nuttallii* (Planch.) St. John, 1920; *Hydrilla verticillata* (Linn. f.) Royle, 1839; *Vallisneria natans* (Lour.) Hara, 1974) coverage from 45% to 70%, being much higher in Gaochun and Jintan than that in other areas (Fig. 3D). The commercial pellet food, trash fish (mainly *Hemiculter leucisculus* Basilewsky, 1855), maize seed and soybean were usually used as feed in mitten crab farming according to all farmers included in the survey (Fig. 3E). The total feed ranged from 9 to $13.8 \times 10^3$ kg ha$^{-1}$ and the ratio of trash fish to total feed varied from 53% to 78% in Gaochun, Jintan and Suzhou area, being much higher than that in other areas (Fig. 3F). The mitten crab yields varied between 1 and $2.7 \times 10^3$ kg ha$^{-1}$ cycle$^{-1}$ among the sites with an average individual body weight of 145 g in Gaochun and Jintan area (Figs. 3G and 3H).

## Growth performance

Almost all crab growth and tissue indices showed significant difference between sites and sex (Table 1). The body weight ($F = 316.14$, $p < 0.0001$, $R^2 = 0.905$), muscle weight ($F = 126.62$, $p < 0.0001$, $R^2 = 0.875$), CL ($F = 42.21$, $p < 0.0001$, $R^2 = 0.681$), CW ($F = 91.90$, $p < 0.0001$, $R^2 = 0.769$), and CF ($F = 47.64$, $p < 0.0001$, $R^2 = 0.549$) of the male crabs was significantly higher, GSI% ($F = 783.29$, $p < 0.0001$, $R^2 = 0.939$) and GH ratio ($F = 491.89$, $p < 0.0001$, $R^2 = 0.913$) were significantly lower than those of female crabs (Table 2). The mean body weight ($F = 25.67$, $p < 0.0001$, $R^2 = 0.905$), muscle weight ($F = 31.19$, $p < 0.0001$, $R^2 = 0.875$), CL ($F = 10.46$, $p < 0.0001$, $R^2 = 0.681$), CW ($F = 11.87$, $p < 0.0001$, $R^2 = 0.769$), and GH ratio ($F = 5.89$, $p < 0.0001$, $R^2 = 0.913$) were significantly higher in Jintan and Gaochun than in the other sites. HSI% ($F = 6.56$, $p < 0.0001$,

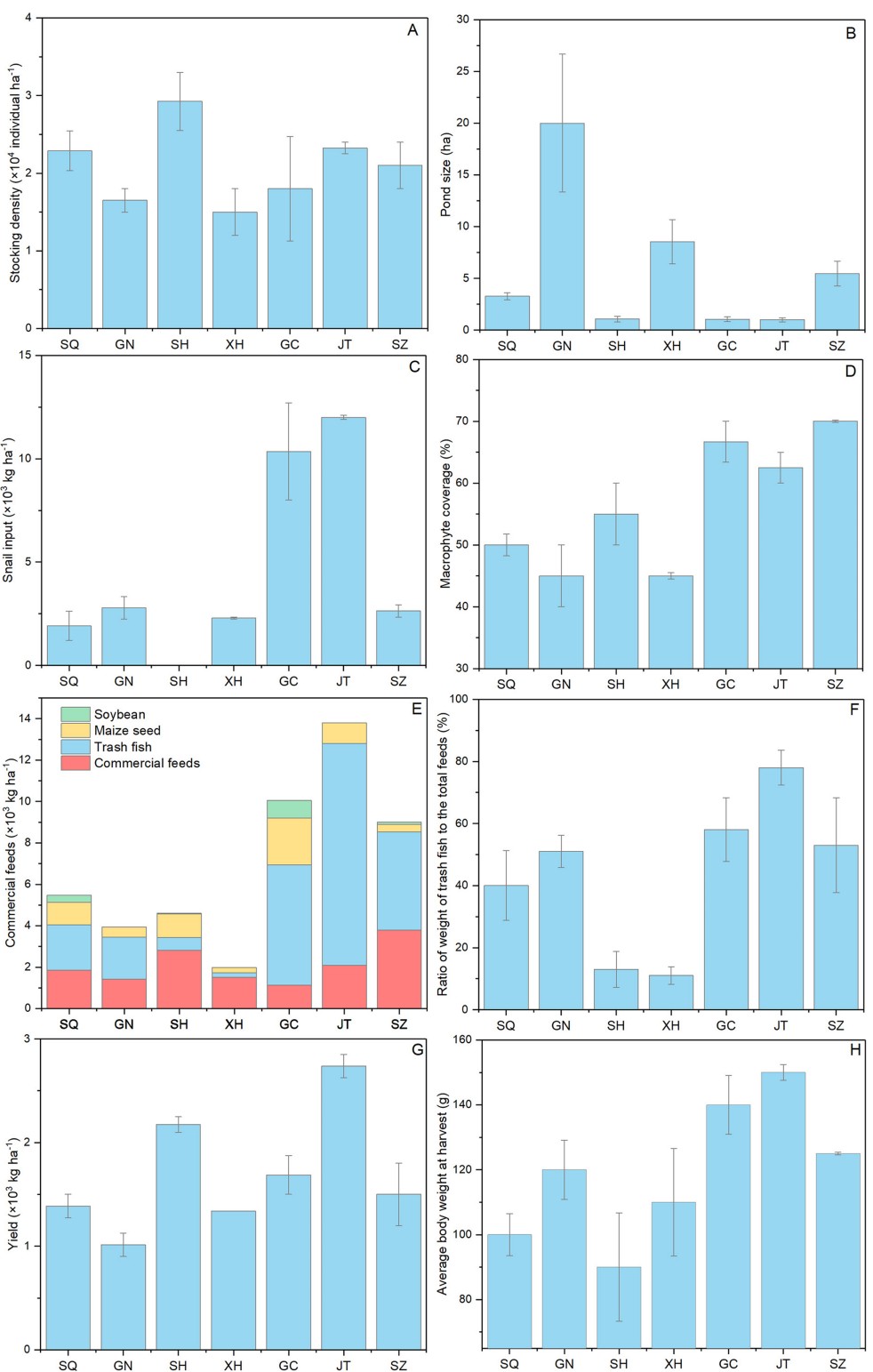

**Figure 3 Farming practice differences of Chinese mitten crab in Jiangsu Province.** SQ, Suqian; GN, Guannan; SH, Sihong; XH, Xinghua; GC, Gaochun; JT, Jintan; SZ, Suzhou.

**Table 1 Growth and tissue indices of Chinese mitten crab in Jiangsu Province (means ± SE).**

| | Suqian ♂ | Suqian ♀ | Guannan ♂ | Guannan ♀ | Sihong ♂ | Sihong ♀ | Xinghua ♂ | Xinghua ♀ | Gaochun ♂ | Gaochun ♀ | Jintan ♂ | Jintan ♀ | Suzhou ♂ | Suzhou ♀ |
|---|---|---|---|---|---|---|---|---|---|---|---|---|---|---|
| Body weight g | 135.34 ± 4.66c | 104.46 ± 8.14c | 137.45 ± 5.37d | 73.38 ± 2.27d | 128.96 ± 5.73c | 97.42 ± 3.06c | 155.26 ± 1.95c | 80.94 ± 1.63c | 151.34 ± 6.93b | 123.4 ± 8.95b | 198.62 ± 4.66a | 125.08 ± 2.92a | 147.64 ± 4.83c | 103.58 ± 3.78c |
| Muscle weight g | 19.26 ± 1.07d | 17.1 ± 2.60d | 19.30 ± 0.93ed | 11.5 ± 1.43ed | 19.68 ± 0.65d | 17.3 ± 0.66d | 20.94 ± 1.6d | 13.12 ± 1.06d | 30.64 ± 1.75b | 19.44 ± 1.1b | 43.06 ± 1.68a | 21.34 ± 0.92a | 26.60 ± 2.08c | 17.52 ± 1.42c |
| CL cm | 5.84 ± 0.19c | 5.70 ± 0.22c | 6.10 ± 0.07c | 5.25 ± 0.05c | 5.96 ± 0.08c | 5.72 ± 0.07c | 6.22 ± 0.07c | 5.38 ± 0.04c | 6.30 ± 0.23b | 6.0 ± 0.27b | 6.90 ± 0.05a | 6.31 ± 0.07a | 6.26 ± 0.07bc | 5.82 ± 0.1bc |
| CW cm | 6.36 ± 0.16c | 6.1 ± 0.17c | 6.55 ± 0.06cd | 5.53 ± 0.05cd | 6.37 ± 0.10c | 6.08 ± 0.13c | 6.7 ± 0.09c | 5.68 ± 0.08c | 6.92 ± 0.19b | 6.2 ± 0.25b | 7.42 ± 0.02a | 6.55 ± 0.05a | 6.66 ± 0.06bc | 6.20 ± 0.08bc |
| HSI% | 8.22 ± 0.37ab | 8.27 ± 0.49ab | 8.14 ± 0.41a | 9.42 ± 0.63a | 7.37 ± 0.39b | 7.00 ± 0.46b | 8.66 ± 0.40a | 9.12 ± 0.47a | 6.65 ± 0.4b | 8.25 ± 0.59b | 7.58 ± 0.45b | 7.25 ± 0.28b | 6.67 ± 0.41b | 7.06 ± 0.33b |
| GSI% | 2.95 ± 0.16bc | 8.41 ± 0.15bc | 2.45 ± 0.28bc | 8.91 ± 0.93bc | 2.91 ± 0.25ab | 8.89 ± 0.39ab | 2.97 ± 0.22ab | 8.93 ± 0.23ab | 2.16 ± 0.18a | 11.67 ± 1.05a | 2.67 ± 0.29ab | 10.99 ± 0.59ab | 2.61 ± 0.51abc | 10.28 ± 0.35abc |
| GH ratio | 0.36 ± 0.03bc | 1.03 ± 0.04bc | 0.30 ± 0.03c | 0.97 ± 0.14c | 0.39 ± 0.03ab | 1.29 ± 0.10ab | 0.35 ± 0.04c | 0.99 ± 0.08c | 0.33 ± 0.02a | 1.42 ± 0.09a | 0.36 ± 0.11a | 1.53 ± 0.11a | 0.41 ± 0.09a | 1.47 ± 0.09a |
| CF | 0.69 ± 0.06a | 0.56 ± 0.02a | 0.60 ± 0.01ab | 0.51 ± 0.00ab | 0.61 ± 0.03ab | 0.52 ± 0.01ab | 0.65 ± 0.02ab | 0.52 ± 0.01ab | 0.61 ± 0.04a | 0.58 ± 0.04a | 0.60 ± 0.02ab | 0.50 ± 0.02ab | 0.60 ± 0.01ab | 0.53 ± 0.02ab |

**Notes:**

Different letters represent significant differences among sites for *p* < 0.05.

CL, carapace length; CW carapace width; HSI, hepatopancreas index; GSI, gonadosomatic index; GH ratio, the ratio of gonad to hepatopancreas; CF, condition factor.

**Table 2 Main and interactive effects of sampling sites and sex on body weight, muscle, CL, CW, HSI %, GSI%, GH ratio and CF.**

| Measurement | Sampling sites | | sex | | Interaction | | $R^2$ |
|---|---|---|---|---|---|---|---|
| | $F(6,12)$ | $p$ | $F(1,12)$ | $p$ | $F(5,12)$ | $p$ | |
| Body weight | 25.67 | 0.000 | 316.14 | 0.000 | 8.04 | 0.000 | 0.905 |
| Muscle weight | 31.19 | 0.000 | 126.62 | 0.000 | 10.28 | 0.000 | 0.875 |
| CL | 10.46 | 0.000 | 42.21 | 0.000 | 2.02 | 0.079 | 0.681 |
| CW | 11.87 | 0.000 | 91.90 | 0.000 | 3.19 | 0.009 | 0.769 |
| HSI% | 6.56 | 0.000 | 3.48 | 0.067 | 1.49 | 0.200 | 0.488 |
| GSI% | 2.49 | 0.034 | 783.29 | 0.000 | 5.12 | 0.000 | 0.939 |
| GH ratio | 5.89 | 0.000 | 491.89 | 0.000 | 4.55 | 0.001 | 0.913 |
| CF | 2.19 | 0.058 | 47.64 | 0.000 | 0.79 | 0.584 | 0.549 |

Note:
CL, carapace length; CW carapace width; HSI, hepatopancreas index; GSI, gonadosomatic index; GH ratio, the ratio of gonad to hepatopancreas; CF, condition factor.

$R^2$ = 0.488) was significantly higher in Xinghua and Guannan than in the other sites (Tables 1 and 2).

## Amino acid composition

Seventeen amino acids were detected including seven essential amino acids in muscles, hepatopancreas and gonads of mitten crab. Muscles contained a higher total amnio acids (TAA), essential amnio acids (EAA), delicious amnio acids (DAA) and non-essential amino acids (NEAA), but had lower EAA/TAA ratio compared to gonads and hepatopancreas (Table 3). Generally, the content of Glutamic acid (Glu) was the highest (muscle 11.12 ± 0.14 mg g$^{-1}$, hepatopancreas 1.32 ± 0.05 mg g$^{-1}$, gonads 7.28 ± 0.14 mg g$^{-1}$) followed by Aspartic acid (Asp) and Alanine (Ala) in all the three tissues. The content in the edible tissues of crabs varied significantly among locations and sex. In muscle, males had more Glu, Glycine (Gly) and Cystine (Cys) than females, and the contents of EAA in Gaochun, Jintan, Suzhou and Guannan were significantly higher than those in Suqian, Sihong and Xinghua (Tables 4 and 5, $F$ = 7.49, $p$ < 0.001, $R^2$ = 0.844). For hepatopancreas, male crabs usually had higher concentration of amino acids than females (Table S1). The concentrations of TAA ($F$ = 11.67, $p$ < 0.0001, $R^2$ = 0.905), EAA ($F$ = 10.44, $p$ < 0.0001, $R^2$ = 0.895), DAA ($F$ = 17.33, $p$ < 0.0001, $R^2$ = 0.934) and NEAA ($F$ = 12.10, $p$ < 0.0001, $R^2$ = 0.909) in Suzhou and Guannan were significantly higher than in the other locations (Table S1; Table 6). Largest differences were found in gonads and related to sex, as the contents of amino acids except Serine (Ser) ($F$ = 1.91, $p$ = 0.188, $R^2$ = 0.213) and Gly ($F$ = 1.27, $p$ = 0.279, $R^2$ = 0.698) were all significantly higher in males than in females (Tables S2; Table 7).

## Nutritional evaluation

All tissues had a high score for all essential amino acids (Table 8). The AAS and CS in hepatopancreas had the highest scores, followed in decreasing order by gonads and muscles. Hepatopancreas and muscles of mitten crabs in Gaochun showed an overall

**Table 3 Amino acid concentration (g 100 g⁻¹ of dry weight) comparison between tissues of crabs from location and sexes (means ± SE).**

|  | Muscle | Hepatopancreas | Gonads | p |
|---|---|---|---|---|
| Asp | 7.2 ± 0.1a | 1.11 ± 0.04b | 5.98 ± 0.31a | 0.000 |
| Thr | 3.5 ± 0.03a | 0.61 ± 0.03b | 4.11 ± 0.30a | 0.000 |
| Ser | 2.93 ± 0.0a | 0.47 ± 0.02b | 3.01 ± 0.04a | 0.000 |
| Glu | 11.12 ± 0.14a | 1.32 ± 0.05c | 7.28 ± 0.14b | 0.000 |
| Gly | 4.88 ± 0.09a | 0.61 ± 0.03c | 2.27 ± 0.03b | 0.000 |
| Ala | 6.63 ± 0.11a | 0.80 ± 0.04c | 4.02 ± 0.18b | 0.000 |
| Cys | 1.27 ± 0.03a | 0.23 ± 0.02b | 1.26 ± 0.09a | 0.000 |
| Val | 3.15 ± 0.04a | 0.58 ± 0.02c | 2.60 ± 0.16b | 0.000 |
| Met | 1.95 ± 0.03a | 0.26 ± 0.01c | 1.07 ± 0.15b | 0.000 |
| Ile | 3.25 ± 0.04a | 0.47 ± 0.02c | 2.65 ± 0.03b | 0.000 |
| Leu | 5.53 ± 0.07a | 0.82 ± 0.03c | 4.09 ± 0.05b | 0.000 |
| Tyr | 2.95 ± 0.03a | 0.36 ± 0.03c | 2.08 ± 0.06b | 0.000 |
| Phe | 3.14 ± 0.04a | 0.56 ± 0.03c | 2.35 ± 0.07b | 0.000 |
| Lys | 5.60 ± 0.07a | 0.71 ± 0.03c | 3.11 ± 0.09b | 0.000 |
| His | 1.58 ± 0.02a | 0.27 ± 0.01c | 1.25 ± 0.05b | 0.000 |
| Arg | 6.89 ± 0.13a | 0.78 ± 0.04c | 3.12 ± 0.18b | 0.000 |
| Pro | 3.56 ± 0.14a | 0.57 ± 0.04b | 4.98 ± 0.73a | 0.000 |
| TAA | 75.09 ± 0.43a | 10.52 ± 0.47c | 55.20 ± 1.09b | 0.000 |
| EAA | 26.12 ± 0.21a | 4.00 ± 0.17c | 19.97 ± 0.20b | 0.000 |
| NEAA | 48.97 ± 0.32a | 6.52 ± 0.30c | 35.23 ± 1.21b | 0.000 |
| DAA | 29.80 ± 0.19a | 3.84 ± 0.17c | 19.54 ± 0.61b | 0.000 |
| EAA/TAA | 0.35b | 0.38a | 0.37 ± 0.01a | 0.000 |

Notes:
Different letters represent significant differences among sites for $p < 0.05$.
Asp, Aspartic acid; Thr, Threonine; Ser, Serine; Glu, Glutamic acid; Gly, Glycine; Ala, Alanine; Cys, Cystine; Val, Valine; Met, Methionine; Ile, Isoleucine; Leu, Leucine; Tyr, Tyrosine; Phe, Phenylalanine; Lys, Lysine; His, Histidine; Arg, Arginine; Pro, Proline. TAA, total amino acids; EAA, essential amino acids; NEAA, non-essential amino acids; DAA, delicious amino acids; EAA/TAA, essential amino acids/total amino acids.

higher AAS and CS scores than that of other sites crabs. All the results were the similar for the EAAI of mitten crabs.

## Drivers of amino acids profile under different farming practices

The ordination between farming practices and sampling sites is shown in Fig. 4A. The two principal components explained 84.3% of the variation. Amino acids contents in site Gaochun, Jintan and Suzhou were placed in the first and fourth quadrants, opposite to other sites, indicating a similar amino acid structure at these sites (Fig. 4A). The RDA analysis revealed that the amino acid content and structure at site Gaochun, Jintan and Suzhou were best explained by snail input, trash fish ratio to the total feed, macrophytes coverage and total trash fish input. All the indices pointed in the same direction, while the stocking density and commercial feed showed a reversed position (Fig. 4A). As illustrated in Fig. 4B, Glu content was best explained by trash fish input, trash fish ratio

**Table 4  Amino acid concentration (g 100 g$^{-1}$ of dry weight) of mitten crab muscle from each site and sex (mean ± SE).**

| | Suqian ♂ | Suqian ♀ | Guannan ♂ | Guannan ♀ | Sihong ♂ | Sihong ♀ | Xinghua ♂ | Xinghua ♀ | Gaochun ♂ | Gaochun ♀ | Jintan ♂ | Jintan ♀ | Suzhou ♂ | Suzhou ♀ | Location | Sex | Location × Sex |
|---|---|---|---|---|---|---|---|---|---|---|---|---|---|---|---|---|---|
| Asp | 6.99 ± 0.17ab | 7.3 ± 0.2ab | 7.27 ± 0.02a | 7.3 ± 0.1a | 6.69 ± 0.08b | 7.04 ± 0.04b | 7.1 ± 0.2ab | 7.23 ± 0.05ab | 7.44 ± 0.04a | 7.2 ± 0.2a | 7.32 ± 0.03a | 7.24 ± 0.02a | 7.16 ± 0.03ab | 7.05 ± 0.02ab | 0.017 | 0.366 | 0.142 |
| Thr | 3.47 ± 0.07a | 3.56 ± 0.04a | 3.52 ± 0.03a | 3.51 ± 0.04a | 3.24 ± 0.04b | 3.43 ± 0.04b | 3.46 ± 0.05a | 3.56 ± 0.01a | 3.74 ± 0.03a | 3.51 ± 0.03a | 3.59 ± 0.02a | 3.6 ± 0.03a | 3.52 ± 0.02a | 3.54 ± 0.04a | 0.000 | 0.248 | 0.003 |
| Ser | 2.9 ± 0.3 | 2.92 ± 0.02 | 2.91 ± 0.03 | 2.91 ± 0.03 | 2.73 ± 0.03 | 2.85 ± 0.04 | 2.89 ± 0.01 | 2.95 ± 0.03 | 3.18 ± 0.03 | 2.85 ± 0.02 | 3.08 ± 0.02 | 2.98 ± 0.02 | 2.96 ± 0.05 | 2.95 ± 0.02 | 0.176 | 0.462 | 0.25 |
| Glu | 10.85 ± 0.29b | 10.84 ± 0.02b | 10.75 ± 0.35b | 10.85 ± 0.3b | 10.45 ± 0.03b | 10.68 ± 0.02b | 10.95 ± 0.04b | 10.83 ± 0.06b | 12.11 ± 0.03a | 10.92 ± 0.02a | 11.65 ± 0.05a | 11.27 ± 0.03a | 12.06 ± 0.03a | 11.42 ± 0.02a | 0.000 | 0.003 | 0.004 |
| Gly | 4.69 ± 0.26c | 4.67 ± 0.06c | 5.01 ± 0.19a | 4.99 ± 0.32a | 5.19 ± 0.04a | 5.13 ± 0.03a | 5.12 ± 0.02b | 4.6 ± 0.1b | 4.56 ± 0.03c | 4.05 ± 0.02c | 5.33 ± 0.03b | 4.76 ± 0.02b | 5.24 ± 0.04a | 5.03 ± 0.03a | 0.000 | 0.001 | 0.141 |
| Ala | 6.52 ± 0.02c | 6.45 ± 0.03c | 7.01 ± 0.09bc | 6.48 ± 0.14bc | 7.2 ± 0.2a | 7.32 ± 0.04a | 6.47 ± 0.03b | 7.37 ± 0.07b | 6.61 ± 0.04cd | 6.12 ± 0.02cd | 6.54 ± 0.02cd | 6.3 ± 0.05cd | 6.28 ± 0.06d | 6.13 ± 0.03d | 0.000 | 0.139 | 0.000 |
| Cys | 1.27 ± 0.06ab | 1.29 ± 0.07ab | 1.27 ± 0.07ab | 1.35 ± 0.05ab | 1.27 ± 0.05a | 1.41 ± 0.01a | 1.34 ± 0.02a | 1.31 ± 0.06a | 1.13 ± 0.04ab | 1.36 ± 0.03ab | 1.01 ± 0.01ab | 1.38 ± 0.01ab | 1.22 ± 0.04b | 1.13 ± 0.03b | 0.012 | 0.001 | 0.002 |
| Val | 2.91 ± 0.03b | 3.22 ± 0.02b | 3.12 ± 0.04b | 3 ± 0.15b | 3.02 ± 0.02b | 2.98 ± 0.04b | 3.11 ± 0.04b | 2.99 ± 0.1b | 3.42 ± 0.03a | 3.28 ± 0.01a | 3.3 ± 0.1ab | 3.26 ± 0.01ab | 3.27 ± 0.07ab | 3.24 ± 0.04ab | 0.000 | 0.465 | 0.046 |
| Met | 2.05 ± 0.05a | 2.09 ± 0.09a | 1.99 ± 0.07ab | 1.88 ± 0.09ab | 1.83 ± 0.03b | 1.89 ± 0.06b | 2.1 ± 0.1a | 2.03 ± 0.03a | 2.03 ± 0.04ab | 1.84 ± 0.01ab | 1.91 ± 0.03ab | 1.88 ± 0.04ab | 1.86 ± 0.03ab | 1.9 ± 0.05ab | 0.01 | 0.248 | 0.327 |
| Ile | 3.15 ± 0.04bc | 3.26 ± 0.05bc | 3.25 ± 0.05bc | 3.19 ± 0.06bc | 3.03 ± 0.01c | 3.16 ± 0.02c | 3.06 ± 0.02c | 3.16 ± 0.03c | 3.52 ± 0.04a | 3.34 ± 0.02a | 3.41 ± 0.03a | 3.38 ± 0.03a | 3.28 ± 0.06b | 3.24 ± 0.02b | 0.000 | 0.001 | 0.007 |
| Leu | 5.31 ± 0.03c | 5.51 ± 0.01c | 5.43 ± 0.06c | 5.46 ± 0.02c | 5.11 ± 0.01d | 5.24 ± 0.04d | 5.33 ± 0.03c | 5.45 ± 0.04c | 6.06 ± 0.02a | 5.68 ± 0.04a | 5.81 ± 0.03a | 5.74 ± 0.04a | 5.61 ± 0.01b | 5.63 ± 0.03b | 0.000 | 0.676 | 0.000 |
| Tyr | 2.93 ± 0.04ab | 3.02 ± 0.02ab | 2.92 ± 0.03ab | 3.09 ± 0.01ab | 2.73 ± 0.02c | 2.77 ± 0.04c | 2.88 ± 0.07b | 2.98 ± 0.08b | 3.12 ± 0.01a | 3.06 ± 0.01a | 3.12 ± 0.04a | 2.93 ± 0.03a | 2.9 ± 0.02b | 2.88 ± 0.08b | 0.000 | 0.388 | 0.008 |
| Phe | 2.94 ± 0.08b | 3.03 ± 0.03b | 3.16 ± 0.03a | 3.32 ± 0.03a | 2.93 ± 0.02b | 3.03 ± 0.02b | 3.09 ± 0.08b | 2.98 ± 0.04b | 3.39 ± 0.02a | 3.18 ± 0.02a | 3.17 ± 0.02a | 3.23 ± 0.03a | 3.24 ± 0.02a | 3.21 ± 0.07a | 0.000 | 0.708 | 0.007 |
| Lys | 5.62 ± 0.07a | 5.77 ± 0.05a | 5.9 ± 0.4a | 5.77 ± 0.06a | 5.39 ± 0.04a | 5.64 ± 0.03a | 5.73 ± 0.03a | 5.79 ± 0.04a | 5.96 ± 0.02a | 5.08 ± 0.04a | 5.52 ± 0.05a | 5.39 ± 0.02a | 5.5 ± 0.05b | 5.29 ± 0.09b | 0.015 | 0.062 | 0.006 |
| His | 1.68 ± 0.01a | 1.6 ± 0.05a | 1.66 ± 0.1a | 1.67 ± 0.05a | 1.5 ± 0.1a | 1.58 ± 0.03a | 1.66 ± 0.04a | 1.59 ± 0.04a | 1.47 ± 0.01a | 1.62 ± 0.02a | 1.54 ± 0.04a | 1.58 ± 0.01a | 1.45 ± 0.05b | 1.46 ± 0.06b | 0.026 | 0.509 | 0.423 |
| Arg | 6.95 ± 0.1a | 6.95 ± 0.03a | 7.4 ± 0.3a | 6.95 ± 0.05a | 6.84 ± 0.04a | 7.05 ± 0.02a | 6.83 ± 0.03a | 6.98 ± 0.03a | 7.15 ± 0.02b | 5.29 ± 0.02b | 7.25 ± 0.02a | 6.85 ± 0.03a | 7.13 ± 0.01a | 6.81 ± 0.01a | 0.000 | 0.000 | 0.000 |
| Pro | 3.97a | 3.86 ± 0.06a | 3.9 ± 0.3ab | 3.65 ± 0.05ab | 3.78 ± 0.04ab | 3.94 ± 0.02ab | 4.14 ± 0.04a | 4.03 ± 0.02a | 2.66 ± 0.01c | 2.93 ± 0.03c | 3.32 ± 0.02b | 3.85 ± 0.05b | 2.82 ± 0.04c | 2.97 ± 0.02c | 0.000 | 0.082 | 0.011 |
| TAA | 74.19 ± 1.65 | 75.36 ± 0.85 | 76.48 ± 2.16 | 75.38 ± 1.55 | 72.93 ± 0.8 | 75.15 ± 0.55 | 75.26 ± 0.85 | 75.82 ± 0.84 | 77.55 ± 0.45 | 71.27 ± 0.56 | 76.85 ± 0.56 | 75.61 ± 0.47 | 75.49 ± 0.63 | 73.89 ± 0.66 | 0.329 | 0.124 | 0.023 |
| EAA | 25.44 ± 0.33b | 26.45 ± 0.29b | 26.38 ± 0.68a | 26.14 ± 0.45a | 24.55 ± 0.17b | 25.38 ± 0.25b | 25.87 ± 0.35b | 25.95 ± 0.32b | 28.12 ± 0.17a | 25.89 ± 0.17a | 26.7 ± 0.28a | 26.47 ± 0.2a | 26.28 ± 0.26a | 26.04 ± 0.34a | 0.001 | 0.424 | 0.006 |
| NEAA | 48.75 ± 1.32 | 48.91 ± 0.56 | 50.1 ± 1.48 | 49.24 ± 1.1 | 48.38 ± 0.63 | 49.77 ± 0.3 | 49.39 ± 0.5 | 49.87 ± 0.52 | 49.44 ± 0.28 | 45.38 ± 0.39 | 50.15 ± 0.29 | 49.14 ± 0.27 | 49.22 ± 0.37 | 47.85 ± 0.32 | 0.062 | 0.068 | 0.043 |
| DAA | 29.06 ± 0.74 | 29.26 ± 0.31 | 30.04 ± 0.65 | 29.62 ± 0.86 | 29.53 ± 0.35 | 30.17 ± 0.13 | 29.65 ± 0.29 | 30.03 ± 0.28 | 30.72 ± 0.15 | 28.28 ± 0.26 | 30.83 ± 0.13 | 29.58 ± 0.12 | 30.74 ± 0.16 | 29.64 ± 0.1 | 0.196 | 0.019 | 0.02 |
| EAA/TAA | 0.34 | 0.35 | 0.34 | 0.35 | 0.34 | 0.34 | 0.34 | 0.34 | 0.36 | 0.36 | 0.35 | 0.35 | 0.35 | 0.35 | – | – | – |

**Notes:**

Different letters represent significant differences among sites for $p < 0.05$.

Asp, Aspartic acid; Thr, Threonine; Ser, Serine; Glu, Glutamic acid; Gly, Glycine; Ala, Alanine; Cys, Cystine; Val, Valine; Met, Methionine; Ile, Isoleucine; Leu, Leucine; Tyr, Tyrosine; Phe, Phenylalanine; Lys, Lysine; His, Histidine; Arg, Arginine; Pro, Proline. TAA, total amino acids; EAA, essential amino acids; NEAA, non-essential amino acids; DAA, delicious amino acids; EAA/TAA, essential amino acids/total amino acids.

**Table 5 Main and interactive effects of sampling sites and sex on muscle amino acid concentration of mitten crab.**

| Measurement | Sampling sites | | sex | | Interaction | | $R^2$ |
|---|---|---|---|---|---|---|---|
| | $F(6,13)$ | $p$ | $F(1,13)$ | $p$ | $F(6,13)$ | $p$ | |
| Asp | 3.91 | 0.017 | 0.87 | 0.366 | 1.95 | 0.142 | 0.720 |
| Thr | 11.98 | 0.000 | 1.452 | 0.248 | 6.07 | 0.003 | 0.887 |
| Ser | 1.78 | 0.176 | 0.57 | 0.462 | 1.50 | 0.250 | 0.591 |
| Glu | 18.02 | 0.000 | 13.05 | 0.003 | 5.54 | 0.004 | 0.917 |
| Gly | 11.36 | 0.000 | 15.97 | 0.001 | 1.96 | 0.141 | 0.873 |
| Ala | 44.07 | 0.000 | 2.47 | 0.139 | 19.04 | 0.000 | 0.965 |
| Cys | 4.29 | 0.012 | 19.20 | 0.001 | 6.53 | 0.002 | 0.857 |
| Val | 9.43 | 0.000 | 0.56 | 0.465 | 2.93 | 0.046 | 0.842 |
| Met | 4.42 | 0.010 | 1.45 | 0.248 | 1.28 | 0.327 | 0.718 |
| Ile | 25.06 | 0.000 | 0.05 | 0.829 | 4.85 | 0.007 | 0.928 |
| Leu | 119.52 | 0.000 | 0.18 | 0.676 | 18.87 | 0.000 | 0.983 |
| Tyr | 15.96 | 0.000 | 0.79 | 0.388 | 4.73 | 0.008 | 0.899 |
| Phe | 19.42 | 0.000 | 0.15 | 0.708 | 4.93 | 0.007 | 0.913 |
| Lys | 4.03 | 0.015 | 4.11 | 0.062 | 5.01 | 0.006 | 0.807 |
| His | 3.46 | 0.026 | 0.46 | 0.509 | 1.07 | 0.423 | 0.664 |
| Arg | 26.34 | 0.000 | 70.72 | 0.000 | 34.41 | 0.000 | 0.969 |
| Pro | 62.73 | 0.000 | 3.52 | 0.082 | 4.31 | 0.011 | 0.967 |
| TAA | 1.28 | 0.329 | 2.68 | 0.124 | 3.57 | 0.023 | 0.694 |
| EAA | 7.49 | 0.001 | 0.68 | 0.424 | 4.99 | 0.006 | 0.844 |
| NEAA | 2.65 | 0.062 | 3.92 | 0.068 | 2.99 | 0.043 | 0.730 |
| DAA | 1.69 | 0.196 | 7.03 | 0.019 | 3.74 | 0.020 | 0.739 |
| EAA/TAA | 89.87 | 0.000 | 14.99 | 0.002 | 3.27 | 0.032 | 0.976 |

Note:
Asp, Aspartic acid; Thr, Threonine; Ser, Serine; Glu, Glutamic acid; Gly, Glycine; Ala, Alanine; Cys, Cystine; Val, Valine; Met, Methionine; Ile, Isoleucine; Leu, Leucine; Tyr, Tyrosine; Phe, Phenylalanine; Lys, Lysine; His, Histidine; Arg, Arginine; Pro, Proline. TAA, total amino acids; EAA, essential amino acids; NEAA, non-essential amino acids; DAA, delicious amino acids; EAA/TAA, essential amino acids/total amino acids.

to the total feed and the total fish. The EAA, such as Thr, Ile, Leu, Val and Phe, was best explained by the snail input.

NMDS analysis showed separation between location and sex in the amino acids content of muscle and gonad of crabs (Fig. 5). All stress values were less than 0.1, indicating a good fit by using this model (*Clarke, 1993*). In muscles, samples taken in site Guannan, Xinghua, Sihong and Suqian were grouped together, and ANOSIM further verified distinct discrepancies with site Gaochun, Jintan and Suzhou (Fig. 5A, $p$ = 0.0293). In gonad, the amino acids content of crabs in females was significantly different from that in males (Fig. 5C, $p$ = 0.0304).

# DISCUSSION

The mitten crab farming practices differed largely among culture areas in Jiangsu Province, especially between the south and north part (Refer to Fig. 5). Most differences in amino

**Table 6 Main and interactive effects of sampling sites and sex on hepatopancreas amino acid concentration of mitten crab.**

| Measurement | Sampling sites | | sex | | Interaction | | $R^2$ |
|---|---|---|---|---|---|---|---|
| | $F_{(6,13)}$ | $p$ | $F_{(1,13)}$ | $p$ | $F_{(6,13)}$ | $p$ | |
| Asp | 8.34 | 0.001 | 12.06 | 0.004 | 7.28 | 0.001 | 0.883 |
| Thr | 9.86 | 0.000 | 14.07 | 0.002 | 4.90 | 0.007 | 0.880 |
| Ser | 3.01 | 0.042 | 4.93 | 0.043 | 1.84 | 0.163 | 0.708 |
| Glu | 30.77 | 0.000 | 38.56 | 0.000 | 15.23 | 0.000 | 0.957 |
| Gly | 7.12 | 0.001 | 14.13 | 0.002 | 4.78 | 0.007 | 0.859 |
| Ala | 18.18 | 0.000 | 36.64 | 0.000 | 11.53 | 0.000 | 0.939 |
| Cys | 4.59 | 0.009 | 6.57 | 0.023 | 3.20 | 0.034 | 0.792 |
| Val | 7.87 | 0.001 | 8.14 | 0.013 | 6.17 | 0.002 | 0.868 |
| Met | 1.70 | 0.194 | 1.14 | 0.303 | 1.66 | 0.203 | 0.603 |
| Ile | 4.60 | 0.009 | 8.56 | 0.011 | 3.41 | 0.027 | 0.802 |
| Leu | 10.60 | 0.000 | 15.54 | 0.001 | 6.61 | 0.002 | 0.895 |
| Tyr | 11.02 | 0.000 | 25.21 | 0.000 | 8.80 | 0.000 | 0.911 |
| Phe | 6.09 | 0.003 | 6.85 | 0.02 | 5.33 | 0.005 | 0.843 |
| Lys | 14.23 | 0.000 | 14.12 | 0.002 | 8.38 | 0.001 | 0.915 |
| His | 2.57 | 0.068 | 0.27 | 0.614 | 1.36 | 0.297 | 0.630 |
| Arg | 24.12 | 0.000 | 35.93 | 0.000 | 21.29 | 0.000 | 0.957 |
| Pro | 26.41 | 0.000 | 50.99 | 0.000 | 4.34 | 0.011 | 0.944 |
| TAA | 11.67 | 0.000 | 18.98 | 0.001 | 7.51 | 0.001 | 0.905 |
| EAA | 10.44 | 0.000 | 13.35 | 0.003 | 7.13 | 0.001 | 0.895 |
| NEAA | 12.10 | 0.000 | 22.01 | 0.000 | 7.50 | 0.001 | 0.909 |
| DAA | 17.33 | 0.000 | 28.52 | 0.000 | 10.74 | 0.000 | 0.934 |
| EAA/TAA | 4.51 | 0.009 | 12.17 | 0.004 | 1.54 | 0.236 | 0.776 |

Note:
Asp, Aspartic acid; Thr, Threonine; Ser, Serine; Glu, Glutamic acid; Gly, Glycine; Ala, Alanine; Cys, Cystine; Val, Valine; Met, Methionine; Ile, Isoleucine; Leu, Leucine; Tyr, Tyrosine; Phe, Phenylalanine; Lys, Lysine; His, Histidine; Arg, Arginine; Pro, Proline. TAA, total amino acids; EAA, essential amino acids; NEAA, non-essential amino acids; DAA, delicious amino acids; EAA/TAA, essential amino acids/total amino acids.

acid composition were observed between culture areas, between types of tissues and for each tissue also between sexes (Refer to Fig. 5). And the farming practices significantly and strongly affected the amino acids composition (Refer to Fig. 4).

Growth of crabs are depending on the available food resources and the quality of those. Macrophytes play a key role in the mitten crab farming, providing natural food, reducing the nutrient loading from culture, and acting as efficient refuges against cannibalism (Zeng et al., 2018). Therefore, high macrophyte coverage is usually considered to enhance crab yield. Wang et al. (2016a) found that the mitten crab yields in farms with a coverage >30% were significantly larger than with low coverage. Accordingly, we found the highest yield in Jintan ($2.7 \times 10^3$ kg ha$^{-1}$ cycle$^{-1}$), having the highest macrophyte coverage (62.5%) among the surveyed ponds, though we did not find an overall significant correlation between yield and macrophytes coverage (Refer to Figs. 3D and 3G). Commercial pellet feed, trash fish, maize seed and soybean were commonly used as feed in

**Table 7 Main and interactive effects of sampling sites and sex on gonad amino acid concentration of mitten crab.**

| Measurement | Sampling sites | | sex | | Interaction | | $R^2$ |
|---|---|---|---|---|---|---|---|
| | $F(6,13)$ | $p$ | $F(1,13)$ | $p$ | $F(6,13)$ | $p$ | |
| Asp | 5.32 | 0.005 | 24.62 | 0.000 | 3.22 | 0.033 | 0.994 |
| Thr | 0.20 | 0.971 | 143.82 | 0.000 | 0.313 | 0.92 | 0.913 |
| Ser | 0.05 | 0.999 | 1.91 | 0.188 | 0.26 | 0.945 | 0.213 |
| Glu | 17.76 | 0.000 | 641.72 | 0.000 | 1.88 | 0.154 | 0.982 |
| Gly | 3.25 | 0.032 | 1.27 | 0.279 | 1.93 | 0.146 | 0.698 |
| Ala | 17.95 | 0.000 | 21.96 | 0.000 | 29.62 | 0.000 | 0.994 |
| Cys | 1.87 | 0.157 | 342.13 | 0.000 | 3.20 | 0.034 | 0.964 |
| Val | 2.17 | 0.109 | 823.15 | 0.000 | 1.16 | 0.382 | 0.984 |
| Met | 3.92 | 0.016 | 14.86 | 0.000 | 1.05 | 0.436 | 0.990 |
| Ile | 1.06 | 0.429 | 28.23 | 0.000 | 3.00 | 0.042 | 0.79 |
| Leu | 1.51 | 0.246 | 18.66 | 0.001 | 3.36 | 0.029 | 0.774 |
| Tyr | 4.08 | 0.014 | 125.27 | 0.000 | 5.54 | 0.004 | 0.929 |
| Phe | 2.76 | 0.055 | 120.95 | 0.000 | 1.68 | 0.200 | 0.913 |
| Lys | 10.43 | 0.000 | 425.16 | 0.000 | 3.13 | 0.037 | 0.973 |
| His | 0.15 | 0.986 | 6.34 | 0.025 | 0.08 | 0.997 | 0.356 |
| Arg | 7.23 | 0.001 | 12.38 | 0.000 | 2.40 | 0.083 | 0.99 |
| Pro | 69.08 | 0.000 | 38.29 | 0.000 | 65.4 | 0.000 | 0.99 |
| TAA | 0.151 | 0.986 | 219.29 | 0.000 | 3.56 | 0.024 | 0.945 |
| EAA | 0.248 | 0.952 | 17.23 | 0.001 | 1.86 | 0.159 | 0.681 |
| NEAA | 0.167 | 0.982 | 766.70 | 0.000 | 4.18 | 0.013 | 0.983 |
| DAA | 3.29 | 0.031 | 11.35 | 0.000 | 3.77 | 0.019 | 0.988 |
| EAA/TAA | 0.516 | 0.787 | 10.06 | 0.000 | 0.25 | 0.952 | 0.987 |

**Note:**
Asp, Aspartic acid; Thr, Threonine; Ser, Serine; Glu, Glutamic acid; Gly, Glycine; Ala, Alanine; Cys, Cystine; Val, Valine; Met, Methionine; Ile, Isoleucine; Leu, Leucine; Tyr, Tyrosine; Phe, Phenylalanine; Lys, Lysine; His, Histidine; Arg, Arginine; Pro, Proline. TAA, total amino acids; EAA, essential amino acids; NEAA, non-essential amino acids; DAA, delicious amino acids; EAA/TAA, essential amino acids/total amino acids.

our study ponds (Refer to Fig. 3E). The crude protein content in commercial pellet feed was 42.6%, lower than in trash fish (64.2%) (Refer to Table S3). The total feed and the ratio of trash fish to total feed differed greatly among the sampling sites (Refer to Figs. 3C, 3E and 3F). And the highest yield and average individual body weight were observed in Jintan with the highest total feed and trash fish input (Refer to Figs. 3E–3H). *Jin et al. (2013)* found that the weight specific growth rate and protein efficiency ratio of juvenile swimming crab increased significantly when the dietary protein level increased from 31.6% to 50.2%. Trash fish has been shown to improve the feed conversion rate and yield in these crab farming (*Wang et al., 2016a*, b), but excessive use of trash fish might cause environmental problems (*Xu et al., 2020*). Besides, replacing fish meal with plant-based or insect-based protein sources has become popular (*Azarm & Lee, 2014*; *Sharawy, Goda & Hassaan, 2016*), though this may affect amino acid metabolism and growth performance as seen in a study by *Yuan et al. (2019)*, with blunt snout bream (*Megalobrama*

**Table 8 Comparison of AAS, CS, and EAAI in three tissues of mitten crab.**

|  | SQ | | GN | | SH | | XH | | GC | | JT | | SZ | |
|---|---|---|---|---|---|---|---|---|---|---|---|---|---|---|
|  | AAS | CS | AAS | CS | AAS | CS | AAS | CS | AAS | CS | AAS | CS | AAS | CS |
| *Muscle* | | | | | | | | | | | | | | |
| Thr | 1.2 | 0.9 | 1.2 | 0.9 | 1.3 | 1.0 | 1.2 | 0.9 | 1.2 | 1.0 | 1.2 | 0.9 | 1.2 | 0.9 |
| Val | 0.8 | 0.6 | 0.8 | 0.5 | 0.9 | 0.6 | 0.8 | 0.5 | 0.9 | 0.6 | 0.9 | 0.6 | 0.9 | 0.6 |
| Met+Cys | 1.3 | 0.8 | 1.2 | 0.8 | 1.4 | 0.9 | 1.3 | 0.8 | 1.2 | 0.8 | 1.2 | 0.7 | 1.2 | 0.7 |
| Ile | 1.1 | 0.6 | 1.1 | 0.6 | 1.2 | 0.7 | 1.0 | 0.6 | 1.2 | 0.7 | 1.1 | 0.7 | 1.1 | 0.7 |
| Leu | 1.03 | 0.8 | 1.0 | 0.8 | 1.2 | 0.9 | 1.0 | 0.8 | 1.1 | 0.9 | 1.1 | 0.9 | 1.1 | 0.9 |
| Phe+Tyr | 1.3 | 0.8 | 1.4 | 0.8 | 1.5 | 0.9 | 1.3 | 0.8 | 1.4 | 0.9 | 1.4 | 0.8 | 1.4 | 0.8 |
| Lys | 1.4 | 1.2 | 1.4 | 1.2 | 1.6 | 1.4 | 1.4 | 1.2 | 1.4 | 1.2 | 1.3 | 1.1 | 1.3 | 1.1 |
| EAAI | 80.2 | | 79.5 | | 90.8 | | 79.2 | | 83.6 | | 80.2 | | 80.3 | |
| *Hepatopancreas* | | | | | | | | | | | | | | |
| Thr | 1.4 | 1.1 | 1.5 | 1.1 | 1.4 | 1.1 | 1.5 | 1.2 | 1.5 | 1.2 | 1.5 | 1.2 | 1.4 | 1.1 |
| Val | 1.1 | 0.7 | 1.1 | 0.7 | 1.1 | 0.7 | 1.1 | 0.8 | 1.1 | 0.8 | 1.1 | 0.8 | 1.1 | 0.7 |
| Met+Cys | 1.3 | 0.8 | 1.2 | 0.8 | 1.4 | 0.9 | 1.3 | 0.8 | 1.4 | 0.9 | 1.3 | 0.8 | 1.4 | 0.9 |
| Ile | 1.2 | 0.7 | 1.1 | 0.7 | 1.1 | 0.6 | 1.1 | 0.7 | 1.1 | 0.7 | 1.1 | 0.7 | 1.1 | 0.7 |
| Leu | 1.1 | 0.9 | 1.1 | 0.9 | 1.1 | 0.8 | 1.1 | 0.9 | 1.1 | 0.9 | 1.1 | 0.9 | 1.1 | 0.9 |
| Phe+Tyr | 1.6 | 0.9 | 1.4 | 0.9 | 1.4 | 0.9 | 1.4 | 0.8 | 1.4 | 0.9 | 1.4 | 0.8 | 1.5 | 0.9 |
| Lys | 1.3 | 1.1 | 1.3 | 1.1 | 1.2 | 1.1 | 1.1 | 1.0 | 1.2 | 1.1 | 1.2 | 1.0 | 1.2 | 1.1 |
| EAAI | 88.4 | | 86.3 | | 86.4 | | 87.1 | | 89.9 | | 87.3 | | 87.8 | |
| *Gonad* | | | | | | | | | | | | | | |
| Thr | 1.8 | 1.4 | 1.8 | 1.4 | 1.9 | 1.5 | 1.9 | 1.5 | 1.9 | 1.5 | 1.9 | 1.5 | 1.9 | 1.5 |
| Val | 0.9 | 0.6 | 1.0 | 0.7 | 1.0 | 0.7 | 1.0 | 0.7 | 1.0 | 0.7 | 1.0 | 0.7 | 1.0 | 0.7 |
| Met+Cys | 1.2 | 0.8 | 1.3 | 0.8 | 1.2 | 0.8 | 1.2 | 0.8 | 1.2 | 0.7 | 1.2 | 0.8 | 1.2 | 0.8 |
| Ile | 1.2 | 0.7 | 1.2 | 0.7 | 1.2 | 0.7 | 1.2 | 0.7 | 1.2 | 0.7 | 1.2 | 0.7 | 1.2 | 0.7 |
| Leu | 1.0 | 0.8 | 1.1 | 0.8 | 1.1 | 0.9 | 1.1 | 0.9 | 1.1 | 0.8 | 1.1 | 0.9 | 1.1 | 0.9 |
| Phe+Tyr | 1.4 | 0.8 | 1.4 | 0.8 | 1.3 | 0.8 | 1.3 | 0.8 | 1.3 | 0.8 | 1.3 | 0.8 | 1.3 | 0.8 |
| Lys | 1.1 | 0.9 | 1.0 | 0.9 | 1.0 | 0.8 | 1.0 | 0.8 | 1.0 | 0.9 | 1.0 | 0.9 | 1.0 | 0.9 |
| EAAI | 83.6 | | 85.8 | | 84.1 | | 84.1 | | 84.0 | | 85.1 | | 83.8 | |

**Note:**
Thr, Threonine; Cys, Cystine; Val, Valine; Met, Methionine; Ile, Isoleucine; Leu, Leucine; Tyr, Tyrosine; Phe, Phenylalanine; Lys, Lysine; EAAI, essential amino acid index.

*amblycephala* Yih, 1955) where fish meals were replaced with cottonseed meal protein hydrolysate at high level (5% and 7%) affecting the AMPK/SIRT1 pathway and inhibiting TOR signaling pathway.

We found the individual size, body weight and muscle weight of the male crabs was significantly higher than for females, while GSI% and the GH ratio were significantly lower (Refer to Tables 1 and 2). Similar results have been found for other crustaceans (*Barrento et al., 2010*; *Yu et al., 2019*). The hepatopancreas is an important organ for the absorption and storage of nutrients, and can synthesize digestive enzymes for food digestion in crustaceans (*Abol-Munafi et al., 2016*), and higher HIS indicates improved hepatopancreatic development and immunity in crabs. GSI measures the development and
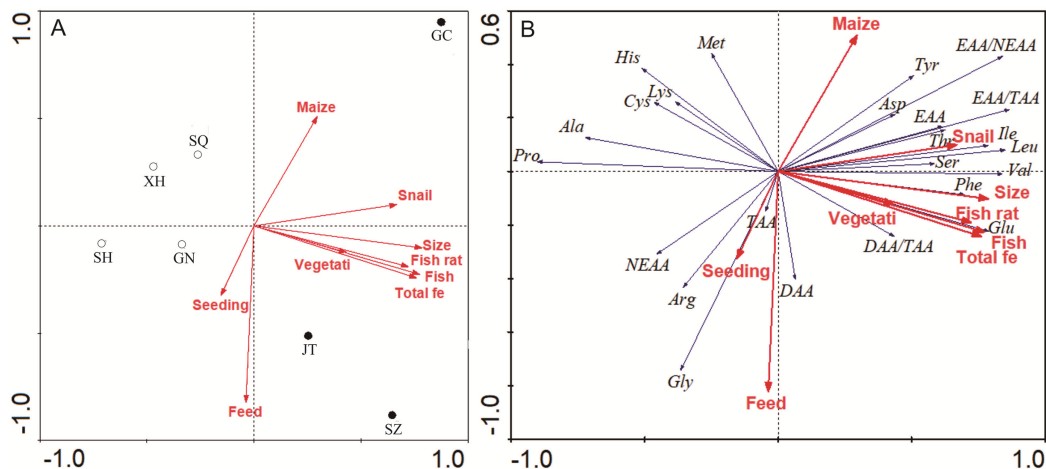

**Figure 4 RDA plots of ordination between farming practices and sampling sites (A), amino acids composition (B).** (A) Ordination between farming practices and sampling sites; (B) Ordination between farming practices and amino acids composition.

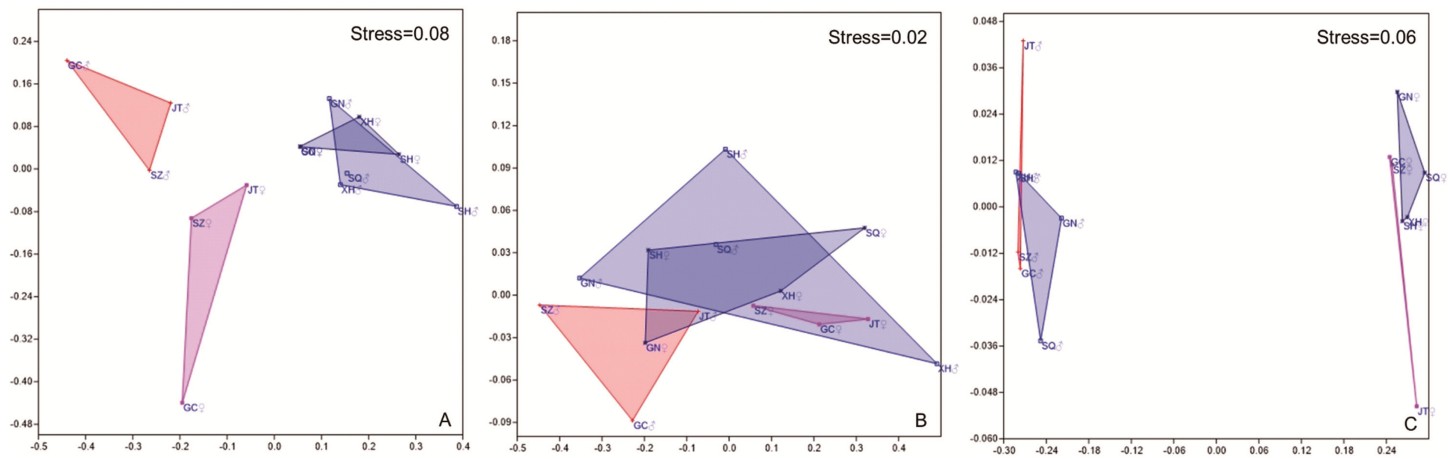

**Figure 5 NMDS results of mitten crab from different sampling sites based on Bray–Curtis similarities calculated from amino acid concentration. A, Muscle; B, hepatopancreas; C, gonad.** GC, Gaochun; SZ, Suzhou; JT, Jintan; XH, Xinghua; GN, Guannan; SH, Sihong; SQ, Suqian. Significant differences in muscle amino acids (A) were found between southern Jiangsu (JT, SZ and GC in pink (♀) and orange (♂) triangle) and northern Jiangsu (GN, XH and SH in gray (♀) and blue (♂) triangle), but gender differences in southern Jiangsu ($p = 0.1024$, JT, SZ and GC in pink (♀) vs. orange (♂) triangle) are not significant, nor in northern Jiangsu ($p = 0.7731$, GN, XH and SH in gray (♀) vs. blue (♂) triangle). Only gender differences were found in gonad amino acids (C) ($p = 0.0289$, males in all sites in orange (♂) and blue (♂) triangle vs. female in all sites in pink (♀) and gray (♀) triangle). Neither sites or gender differences were found in hepatopancreas amino acids (B).

maturation of gonadal tissues relative to body weight (*Wu et al., 2020b*). Data on HIS and GSI are usually impacted by diet protein and living environment. *Muhammad et al. (2019)* found that the high water flow velocities inhibit the production of hepatopancreas and gonad. Addition of glutathione can significantly increase HSI in Chinese mitten crabs (*Liu et al., 2019*). But workers like *Yu et al. (2019)* and *Butts et al. (2020)* reported that diet with different protein did not influence organ-somatic indices (e.g., HSI, GSI). We found lower HIS of crabs in Gaochun and Jintan area, although higher-protein trash fish feed was input in these areas (Refer to Table 1).

Amino acids are the main constituent that have been used in assessing the nutritional value of the fishes (*Azarm & Lee, 2014*; *Hussain et al., 2018*) and crustaceans (*Wang et al., 2018*; *Liu et al., 2019*). Several studies have reported that differences in the amino acid content in the edible tissues of mitten crabs from different sampling sites may largely be attributed to environment and type of food they eat, but not elucidated how environment and food influenced the amino acid composition in detail (*Wu et al., 2020a*; *Liu et al., 2019*). Generally, we found TAA, EAA and DAA in muscle and gonads to be significantly higher than in hepatopancreas (Refer to Tables 3– 7). These results are consistent studies for other crustaceans (*Barrento et al., 2009*; *Barrento et al., 2010*; *Wu et al., 2020a*). EAA are those that either cannot be synthesized or are inadequately synthesized by animals relative to needs (*Akiyama, Oohara & Yamamoto, 1997*). EAA need, therefore, be provided from the diet under conditions where rates of utilization are greater than rates of synthesis. EAA in muscle in Gaochun, Jintan, Suzhou and Guannan (characterized with higher trash fish input), were significantly higher than those in other study sites (Refer to Tables 4 and 5). DAA contribute to the flavor of the crab, which is an appealing flavor in mitten crab farming. Gly and Ala contribute to a sweet taste, whereas Glu and Asp provide a strong umami taste (*Yamaguchi et al., 1971*). In the present study, Glu was the main amino acid contributing to the umami taste in the mitten crabs. The DAA/TAA values were between 0.34 and 0.37 in all the tissues (Refer to Table 4; Tables S1 and S2). It's similar to crustaceans reported in other studies, such as brown crab *Cancer pagurus* (Linné, 1758) (0.31–0.38) (*Barrento et al., 2010*), swimming crab *Portunus trituberculatus* (Miers, 1876) (0.30–0.37) (*He et al., 2017*) and Chinese mitten crab *Eriocheir sinensis* (Milne Edwards, 1853) (0.34–0.39) (*Wang et al., 2018*).

The EAA/TAA ratio, AAS, CS and EAAI are important factors when evaluating the nutritional quality of amino acid for fishery products (*Guo et al., 2014*; *He et al., 2017*; *Wu et al., 2020a*). In our study, the average EAA/TAA values were 0.35 in muscles, 0.38 in hepatopancreas and 0.37 in gonads, which were close to the standard values (0.4) recommended by FAO/WHO (*FAO/WHO, 1991*) (Refer to Table 4; Tables S1 and S2). No significant differences in EAA/TAA values were observed among sampling sites (Refer to Table 4; and Table S1).

The AAS, CS, and EAAI values had the highest scores in the hepatopancreas followed by the muscle and gonads, thus hepatopancreas had the highest nutritional quality in terms of amino acids (Refer to Table 8). Hepatopancreas and muscles of mitten crabs in Gaochun showed overall higher AAS and CS scores than in crabs from the other sites (Refer to Table 8). Except for the Val, the AAS scores were higher than the FAO/WHO standard. The CS values of amino acids in the three parts were lower than the reference egg protein standard, except for Lys in muscles, Thr and Lys in hepatopancreas as well as Thr in gonads (Refer to Table 8). Similar results were observed in *Wu et al. (2020a).* The EAAI of some amino acid more than 100 indicates this amino acid is non-limited and will meet the requirements of children or adults. The EAAI values ranged from 79.2 to 90.6 (average 84.6). Compared with other fishery products, the values were higher than in crabs sampled from Yangchenghu Lake (66.5) (*Guo et al., 2014*), from rice-field (82.2) (*Wu et al., 2020a*), and in roach *Paramisgurnus dabryanus* (Dabry de Thiersant, 1872)

(56.1) (Dong et al., 2018), lower than in swimming crab *Portunus trituberculatus* (Miers, 1876) (146.8) (*He et al., 2017*), and brown crab *Cancer pagurus* (Linné, 1758) (157.8) (*Barrento et al., 2010*). Therefore, all crab sampled showed high nutritional value suitable for serving human dietary needs.

The amino acid content and structure in Gaochun, Jintan and Suzhou were best explained by snail input, trash fish ratio to the total feed, macrophytes coverage and total trash fish supply (Refer to Fig. 4). The environment, seasonal variations, sexual changes and the amount and quality of food are key factors contributing to changes in amino acids content in marine species (*Kasozi et al., 2019*). *Liu et al. (2018)* argued that a more balanced diet with fish, detritus and microalgae and a higher macrophytes coverage could increase the free TAA content in wild mitten crabs, while polluted environments would negatively affect protein efficiency ratio and chemical composition. *Hussain et al. (2018)* reported that loss in the nutritional value in fish harvested from polluted sections of the River Chenab. Dietary protein had a significant effect on whole-body crude protein and muscle amino acid profile (*Yu et al., 2019*). Individual and total EAAs significantly increased when the dietary protein levels increased from 31.6% to 50.2% (*Jin et al., 2013*). Trash fish as a high protein feed is usually operated in the mitten crab culture. In our study, the ratio of trash fish to total feed in ranged from 53% to 78% in the south area of Jiangsu Province (Gaochun, Jintan and Suzhou), being much higher than that in other areas (11–51%) (Refer to Fig. 3F). Accordingly, the crab muscle EAA contents were higher in sites with high trash fish input. In fact, the trash fish as feed in aquaculture have been banned in China in recent years, mainly due to potential environmental pollution. But according to the survey from 156 crab farms around Hongzehu Lake, *Wang et al. (2016b)* did not find significant differences in total nitrogen and total phosphorus loading among feed types. They argued that use of plants and filter-feeding snails could help maintain good water quality.

## CONCLUSION

In order to explain how the farming operations affected the amino acids composition of Chinese mitten crabs, crabs were collected and farming practices were investigated from 7 main farming areas in Jiangsu Province. The results showed that significant differences in farming practices and amino acids content of three edible tissues among the pond-reared male and female mitten crabs across Jiangsu Province. Higher snail supply, macrophytes coverage, total commercial feed, and the ratio of trash fish to total feed input in the south area of Jiangsu Province (Gaochun and Jintan) lead to higher average individual body weight, higher commercial yields and altered the amino acid content and structure. The crabs collected delivered high nutrient value (AAS >1 except Val) with Glu, Asp, and Ala as the main amino acids contained. The muscle EAA contents in Gaochun, Jintan, Suzhou and Guannan were significantly higher those in other sites. The research suggests that the Chinese mitten crab is healthy for human consumption and the trash fish feed and snail would influence the amino acids composition of crabs and taste. Maybe more sample collections could support the results strongly. Further studies should elucidate how the trash fish feed and snail affect the nutrient value and whether

insect source protein can be used as an alternative feed in crab aquaculture based on experimental investigations. Such studies may provide credible guidance to farmers on how to obtain high nutrient values of mitten crab culture and much deeper understanding of how the feeds affect the quality.

## ACKNOWLEDGEMENTS

We thank Anne Mette Poulsen for critical editorial assistance and Prof. Ma Ronghua for map rendering of Jiangsu Province.

### Funding

This work was supported by the National Key Research and Development Program of China (2020YFD0900500), the Natural Science Foundation of China (No. 31972813) and the Jiangsu Province Scientific Research Foundation (TH2018303 & CX(20)2026). Erik Jeppesen was supported by the Tübitak Outstanding Researcher Program (BIDEB 2232 & Project 118C250). The funders had no role in study design, data collection and analysis, decision to publish, or preparation of the manuscript.

### Grant Disclosures

The following grant information was disclosed by the authors:
National Key Research and Development Program of China: 2020YFD0900500.
Natural Science Foundation of China: 31972813.
Jiangsu Province Scientific Research Foundation: TH2018303 and CX(20)2026.
Tübitak Outstanding Researcher Program: BIDEB 2232 & 118C250.

### Competing Interests

The authors declare that they have no completing interests.

### Author Contributions

- Qingfei Zeng conceived and designed the experiments, performed the experiments, analyzed the data, prepared figures and/or tables, and approved the final draft.
- Yuxia Xu performed the experiments, prepared figures and/or tables, and approved the final draft.
- Erik Jeppesen analyzed the data, authored or reviewed drafts of the paper, and approved the final draft.
- Xiaohong Gu analyzed the data, authored or reviewed drafts of the paper, and approved the final draft.
- Zhigang Mao conceived and designed the experiments, performed the experiments, prepared figures and/or tables, and approved the final draft.
- Huihui Chen conceived and designed the experiments, performed the experiments, prepared figures and/or tables, and approved the final draft.

## Data Availability

The data are available in the Supplementary File.

## Supplemental Information

Supplemental information for this article can be found online at http://dx.doi.org/10.7717/peerj.11605#supplemental-information.

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
