# Peer review of "Farming practices affect the amino acid profiles of the aquaculture Chinese mitten crab"

_PeerJ, doi:10.7717/peerj.11605_

## Round 0.1 · original submission · Major Revisions

The reviewers have considered a lot of merit to your work. One of the reviewers recommended acceptance, however, the other two provided comments that in the editor's opinion, will contribute to enhancing your work.

Together with reviewers, the editor adds the following suggestions:

1) Compulsorily, authors should provide at the beginning of materials and methods, an 'overview of the experimental program'. In this section, authors should succinctly relay a snapshot of the entire study, reiterating the objective of the study, supported with a schematic representation, example of the following diagram can be: design of study / divided into two parts where one is >collection of samples (source) > air-freighting of live samples>arrival at laboratory> analytical procedures (showing the parameters and sample replication, etc and the other > data of farming practices >
etc
Authors need to use this platform to demonstrate snapshot for readers to follow

2) Statistical analysis shows a two-way ANOVA was used. Why is there no interaction shown? The authors should provide an ANOVA Table, showing the individual and interactions, of parameters, with the P-, F-, and R-sq. values. Information of this should be relayed in the results, and its discussion, incorporated in the discussion sections.

3) One of the reviewers identified text that seems wordy. It would useful to break down sentences, where possible, and make them simple, and connect them, in simple ways.

4) try to see how to tactfully/strategically reduce the discussion, limiting the repetition of results in its entirety.

5) any future recommendations/ prospects in the conclusions? the direction of future studies should be provided, authors kindly brainstorm on it ok.

It is a brilliant study. Looking forward to your revised manuscript.

Reviewer 1 ·

Basic reporting

ok

Experimental design

ok

Validity of the findings

ok

Additional comments

this paper has good structure and high quality.

Reviewer 2 ·

Basic reporting

General structure of the paper is pretty basic and nothing to complain about, but the flow between the sections can be improved. My main probe with the basic reporting is a poor readability of the text. English grammar is not that problematic, so I would not recommend to show paper to the native speaker (thought its never hurts per se), but what I will recommend is to drastically improve the text. Now text is stuffed with wordy, complex unreadable sentences, which are confusing and not always conveying what authors seems to want to say. I marked numerous examples of that in the annotated pdf, but in short, even experienced authors can always improve their writing. I think that Hotaling (2020) gave excellent tips to do just that. I will strongly recommend to re-write the main text with Hotaling’s tips in mind (https://aslopubs.onlinelibrary.wiley.com/doi/full/10.1002/lol2.10165). I think that paper’s readability, enjoyment of the readers and, eventually, number of citations will gain immensely from trimming overly wordy text into the more concise manuscript.

Experimental design

Experimental design raised my main issue with a manuscript. Authors have only sampled 5 crabs of each sex from each of the sampled ponds. I was surprised that a) it was always 5 (10) crabs per pond without correction of the sampling effort for the pond’s size, even though surface area of the sampled ponds varied from >4ha to over 20ha; b) I was confused by the following statement (lines 94-96) “Since the hepatopancreas and
gonads from a single crab had relatively low total weight, the tissue from 5 crabs of each sex per
pond were pooled to form and at least two replicates were then analysis in each culture area. “ Does authors pooled the tissues from 5 crabs into two batches at least? Than those are not replications, those are pseudoreplications, as you are sampling your tissues from the same pool of 5 animals. Therefore I am a bit concerned about small sample size. I would like authors to address the possible pseudoreplications, small sample size and why was not the amount of the specimens sampled corrected for the surface area of the sampled pond

Validity of the findings

I looked through the analysis and the raw data and the conclusions of the authors seems to be well supported by data. Regrettably, authors did not used open-source software, so checking the code is not an option. I think that description of the statistical outcomes and analysis are fuzzy in some places and language that authors use to talk about stats should be elucidated. Please see pdf annotations for the details. In general, while I find the finding valid and supported by data, my issues with sample size and adjacent parts of the experimental design (which I described above) should be addressed first for the proper evaluation of the results.

Additional comments

I commend this paper for looking into the important issues of the development of sustainable acculturate in one of the largest world economies- China. I think paper asks interesting questions and supports its answers with appropriate data, but considerable efforts should be taken to improve the text’s readability and to better explaining experimental design.
I think paper can be accepted after the major review is implemented.

Annotated reviews are not available for download in order to protect the identity of reviewers who chose to remain anonymous.

Reviewer 3 ·

Basic reporting

In this manuscript, literature references and background information have been provided.

Experimental design

no comment

Validity of the findings

This manuscript has shown the impactful and novel results where the rationale and benefit is clearly stated.

Additional comments

In this paper, the author investigated and observed differences in farming practices and amino acids contents of three edible tissues among the pond-reared mitten crabs across Jiangsu Province. The results showed that the snail supply, macrophytes coverage, total commercial feed and the ratio of trash fish to total feed input have influence to the average individual size and commercial yield of mitten crab. The author also discussed the nutrient value and taste of mitten crabs from different locations (i.e., north and south area of Jiangsu). All the information would be very important and indicative to the farming practices.

Questions:
- [line 119 and 120] the eqn 5 and eqn 6 are the same at the right side.
- About the Table 1, it would be helpful to reformat the Table 1. Part of the information is out of the margin right now.

---

## Round 0.2 · Major Revisions

Authors, thank you for your efforts in revising the manuscript. Upon initial check, the ‘overview of experimental program’ as requested has not been included. Authors appear to have avoided this. Kindly develop this section as requested, and incorporate it, not only in the rebuttal but also in the text itself. A schematic diagram reflecting the flow of the experimental program has to be developed to form a part of this section. It is important to reiterate the importance of this section. This section helps to provide a snapshot of your work and guides the readers on your study design.

All the new Tables should be moved from supplementary into the main work.

Also, all actual statistical values, and not simply p<0.05, including the R-square value, and interactions have to be highlighted in results and referred to in the discussion.

Kindly revise the manuscript to all above, before it can proceed. Look forward to seeing your revision very soon. Thank you.

---

## Round 0.3 · Minor Revisions

Thank you authors for revising your manuscript. Reviewers have recommended acceptance. However, editor requests the authors to attend to the following:

1) Schematic overview of study, provide some description of the figure, reiterating how it relates to the objective of this work. Ofcourse, succinctly mention the stages shown in the figure, make sure it is all directed towards demonstrating the relevance of this work, and why these steps have to be followed to achieve the said objectives.

2) In the discussion, kindly insert (Refer to Table ?? or Fig. ??) at all the places where the results being referred to are discussed. The editor expects to see all the Tables, and all the Figures being referred to in the discussion. Please, the importance of this is to guide readers to know, which Table is being discussed, which Figure is being discussed, at every aspect of the discussion.

3) Please, in the conclusions, reiterated why this study was conducted, and assert the objective succinctly.

Before stating direction for future work, what makes this study relevant? Where the entire objectives achieved? Have you any potential limitations?

Look forward to receiving the revised manuscript. This is a very brilliant work :)

Reviewer 3 ·

Basic reporting

After authors' revision, the manuscript has become clear and unambiguous, professional english used throughput.

Experimental design

The original research aims and scope has been well demonstrated in this manuscript. Research questions are well defined and illustrated.

Validity of the findings

Meaningful data and results have been provided and statistically analyzed. All of them are robust, statistically sound and controlled.

Additional comments

The authors have answered my questions and comments in the revised manuscript. I would like to support this manuscript publish in the PeerJ journal. Thanks.

---

## Round 0.4 · Minor Revisions

Thank you for revising the manuscript. Some areas appear to have been indicated in rebuttal as done but were not done as said in the revised manuscript.

Schematic overview of study
Please move this information from figure to the text.
Line 86 onwards should read:
“Figure 1 shows the schematic overview of the study. Herein, both sample collection and farming practices survey was carried out at the same time when the crabs were harvested. Then the crabs were dissected to three edible tissues of both sexes for further growth performance and amino acid analysis. The drivers of amino acids composition of mitten crabs cultured in different farming patterns using a redundancy analysis (RDA).”

The figure legend should read:

A schematic overview of the study. n=number of sample replicates for amino acidanalysis.

Discussion:
Please, kindly insert 'Refer to...before all figures and tables as mentioned in the discussion

For example, (Refer to Fig. 5) or (Refer to Table S3)

The aim is to guide the reader to follow your work.

Other minor corrections:

Line 235: Replace 'therefore' with 'Besides,'
Line 251: But, workers like Yu et al. (2019) and Butts et al. (2020) reported....
Line 264: please change 'must' to 'need'

---

## Round 0.5 · accepted · Accept

Thank you authors for attending to all the concerns raised, and revising your manuscript. The authors have benefited from the peer-review process. This revised work is now acceptable for publication. Thank you for considering PeerJ as your journal of choice, and counting on your continued future scholarly works. Congratulations